# Heart Failure after Cardiac Surgery: The Role of Halogenated Agents, Myocardial Conditioning and Oxidative Stress

**DOI:** 10.3390/ijms23031360

**Published:** 2022-01-25

**Authors:** José Luis Guerrero-Orriach, Maria Dolores Carmona-Luque, Laura Gonzalez-Alvarez

**Affiliations:** 1Institute of Biomedical Research in Malaga, 29010 Malaga, Spain; 2Department of Anesthesiology, Virgen de la Victoria University Hospital, 29010 Malaga, Spain; 3Department of Pharmacology and Pediatrics, School of Medicine, University of Malaga, 29010 Malaga, Spain; 4Maimonides Institute of Biomedical Research in Cordoba (IMIBIC), 14004 Cordoba, Spain; mariadolores.carmona@imibic.org; 5Cellular Therapy Unit, Reina Sofia University Hospital, 14004 Cordoba, Spain; 6University of Cordoba, 14004 Cordoba, Spain

**Keywords:** heart failure, preconditioning, postconditioning, halogenated, ACDHUVV-16

## Abstract

Heart disease requires a surgical approach sometimes. Cardiac-surgery patients develop heart failure associated with ischemia induced during extracorporeal circulation. This complication could be decreased with anesthetic drugs. The cardioprotective effects of halogenated agents are based on pre- and postconditioning (sevoflurane, desflurane, or isoflurane) compared to intravenous hypnotics (propofol). We tried to put light on the shadows walking through the line of the halogenated anesthetic drugs’ effects in several enzymatic routes and oxidative stress, waiting for the final results of the ACDHUVV-16 clinical trial regarding the genetic modulation of this kind of drugs.

## 1. Introduction

Heart disease is a leading cause of mortality that frequently requires surgery. However, patients who undergo cardiac surgery often develop heart failure, which is a major cause of morbidity and mortality in this setting [1,2]. Heart failure is favored by ischemia induced during extracorporeal circulation (EC), where a cardioplegic solution is infused to cause diastolic cardiac arrest. This complication is associated with the duration of EC and baseline cardiac status of the patient. There is a growing interest in the field of anesthesiology and postoperative critical care regarding the cardioprotective effects of myocardial preconditioning (and postconditioning) induced by halogenated anesthetics (sevoflurane, desflurane, or isoflurane) versus intravenous hypnotics (propofol) [3].

This review sheds a nuanced light on the impairment mechanisms induced by heart dysfunction and the improvement effector and modulator mechanisms induced through the use of anesthetics. To this purpose, we explored the role of the halogenated anesthetic agents regarding the oxidative stress, enzymatic pathways, and gene modulation (mainly through miRNAs).

## 2. Clinical Evidence

Myocardial dysfunction following cardiac surgery is a major cause of morbidity and mortality. Ischemia induced during the procedure may cause cardiac dysfunction and low-cardiac-output syndrome [1,4]. Ischemia is induced through physical procedures (local hypothermia) and the infusion of a cardioplegic solution into the coronary arteries. This solution has a high concentration of potassium ion and diffuses within the coronary circulation and cardiomyocytes, inducing cardiac arrest and decreasing basal metabolic oxygen consumption. This solution can be infused either once or several times, based on its composition and the type of procedure.

Heart failure, when normal cardiac function is restored after induced ischemia, occurs in 20% of cases. This complication may cause a cardiogenic shock, with high mortality rates [1]. At the preoperative level, cardiac dysfunction is associated with preoperative heart status, whereas at intraoperative and postoperative levels, cardiac dysfunction is related to the type of surgery, the duration of ischemia, and the extracorporeal circulation.

During the last 20 years, multiple studies have been conducted to assess the cardioprotective effects of anesthetic agents in cardiac surgery when administered intraoperatively, postoperatively, or both.

### 2.1. Ischemic Pre- and Postconditioning

Surgeons draw on their knowledge regarding ischemic pre- and postconditioning mechanisms to induce cardioprotection using anesthetic agents.

In ischemic myocardial conditioning, the coronary arteries are exposed to brief cycles of clamping and declamping prior to sustained ischemia with the aim to reduce heart damage [5,6]. Multiple studies have assessed the size of ischemic areas (areas of myocardial infarction) in models exposed to the technique described above, whereas, in other studies, ischemia is induced without previous conditioning. The results show that the size of the infarcted area was smaller in the models exposed to ischemic pre-postconditioning [7,8]. Ischemic preconditioning, however, involves a high risk in clinical terms. Determining the optimal duration of ischemia for each patient and achieving cardioprotection without causing cardiac ischemia is challenging.

### 2.2. Anesthetic Pre- and Postconditioning

To overcome the challenges posed by ischemic myocardial conditioning and achieve cardioprotection, some authors recently explored the potential of anesthetic agents to induce pre- and postconditioning. This technique mimics ischemic conditioning but prevents the risk of cardiac ischemia [9].

In the past, during anesthesia induction, hypnotic agents were administered with the only aim of acting on the central nervous system. In recent years, however, owing to their cardioprotective effects, halogenated agents (primarily, sevoflurane, desflurane, and isoflurane) have become the hypnotic agents of choice—versus intravenous hypnotics—in the intra- and postoperative period of cardiac surgery [9,10,11,12,13,14,15,16,17]. Early clinical trials demonstrate differences in heart damage based on cardiac enzyme levels. In clinical terms, differences were also observed in relation to the use of inotropics to treat heart failure of low-cardiac-output syndrome. Thus, the use of halogenated agents for inhalational anesthesia, versus propofol, has been documented to exert cardioprotective effects and reduced myocardial injury [9,14]. In a later study, De Hert et al. documented an association between the duration of infusion and the concentration of the halogenated agent and its cardioprotective effects [14,15,16]. In contrast, recent studies question the morbidity and short-term mortality results reported for this group of patients in relation to the type of anesthetic administered [17], but Bonani et al. observed that volatile anesthetics were superior to propofol with regard to long-term mortality in cardiac surgery with cardiopulmonary bypass [18].

In our opinion, it is very difficult that a single technique can help to overcome all the challenges posed by cardiac surgery and improve morbidity and mortality, regardless of the type of procedure. Hence, the mere possibility that a drug may improve cardiac function and reduce perioperative heart failure in cardiac surgery strongly supports its use.

### 2.3. Optimizing the Clinical Use of Halogenated Agents

Based on the findings described above, several studies have been conducted to determine whether the sustained use of halogenated agents in the immediate postoperative period exerts similar cardioprotective effects as the ones observed when administered only intraoperatively, with inconsistent results [19,20,21,22].

Steurer et al. [20] report a reduction in myocardial injury in cardiac-surgery patients who received a halogenated agent both intraoperatively and postoperatively (instead of propofol). The authors postulated that cardioprotection could have been achieved through late preconditioning and postconditioning mechanisms. Conversely, Hellström et al. [21] found no significant differences between treatment groups. It is worth mentioning that the postoperative period of sedation was extremely short in this study and could not exert the beneficial effects documented in other studies. These works assessed early late anesthetic preconditioning and/or postconditioning in cardiac-surgery patients. These phenomena are directly related to the immediate postoperative hours.

Promising results have been obtained regarding myocardial conditioning induced by halogenated agents (sevoflurane) when administered intraoperatively and maintained during the immediate postoperative period, as compared to the use of an intravenous anesthetic (propofol). The sustained administration of the halogenated agent within the first six postoperative hours seems to be determinant for achieving cardioprotection [22,23,24].

### 2.4. Our Experience

Our research group has undertaken numerous studies in this line of research. Firstly, we explored differences between intraoperative and postoperative anesthesia-sedation and propofol (agent of reference for maintenance of intravenous anesthesia), as compared to patients who received sevoflurane in the two periods. This first work revealed that halogenated agents exerted beneficial effects regarding propofol. This conclusion was based on the differences observed in biochemical markers of myocardial injury, heart dysfunction, and inflammatory response. This study, however, had some limitations in relation to the beneficial effects of the intraoperative use of sevoflurane; the study did not involve a control group that only received sevoflurane intraoperatively and administered another sedative postoperatively. This means that the beneficial effects obtained cannot be directly credited to the intraoperative use of sevoflurane [22].

A second study was then performed to assess the potential cardioprotective effects of halogenated agents. This study involved three groups, one of which received sevoflurane intraoperatively and propofol postoperatively. The results demonstrated a reduction in the levels of biochemical biomarkers of myocardial injury and cardiac dysfunction (Troponin I, NT-ProBNP), with differences across the three groups. These biomarkers were higher in the group that received propofol intra- and postoperatively as an anesthetic. Notably, there were statistically significant differences between the other two groups, suggesting that cardioprotection improved when sevoflurane was administered both intra- and postoperatively [23].

In a later study based on the same methods, we explored the potential effector mechanisms of cardioprotection exerted by halogenated agents. A clinical trial was conducted involving two groups of patients undergoing revascularization surgery. We sought to explore the enzymatic mechanisms by which the intra- and postoperative use of sevoflurane exerts cardioprotection. The first group received sevoflurane intra- and postoperatively (SS group); the second group was administered sevoflurane intraoperatively and propofol for postoperative sedation (SP group); and the third group received propofol intra- and postoperatively (PP group). The results revealed that cardioprotection induced by sevoflurane was mediated by the overexpression of the enzymes that regulate drug-induced myocardial pre- and postconditioning [24].

The incidence of myocardial injury (assessed based on plasma levels of troponin I) was lower in the group that received sevoflurane intra- and postoperatively. This effect is primarily related to the elevation of the enzymes involved in the RISK and SAFE (Akt/ERK 1⁄2/STAT5) pathways, which causes a decrease in the expression of markers of cellular apoptosis (caspase3). In clinical terms, these effects reduced cardiac dysfunction (LCOS), which helped preserve kidney function.

Gene modulation regulates the expression of some of the key enzymes involved in cardioprotection exerted by halogenic agents, such as Akt, ERK 1⁄2, and STAT proteins, all related to the SAFE and RISK pathways. These findings encouraged our group to perform further studies, currently on course. In these studies, we postulate that anesthetics would induce myocardial conditioning through gene modulation, thereby regulating enzymatic mechanisms.

Non-codifying RNA and, more specifically, miRNAs, is one of the most important subjects of study in the field of diagnostics and therapeutics [24]. Our group recently published a study reporting the preliminary results of the ACDHUVV-16 trial. In this study, we performed the NGS sequencing of the miRNAs of cardiac-surgery patients. Patients were allocated to two groups based on the intra- and postoperative anesthetic administered (sevoflurane vs. propofol). The results identified several miRNAs as mediators of cardioprotection. However, these preliminary results must be confirmed by the results of the study, which will presumably be obtained by 2022 [25].

The possibility that anesthetics induce gene modulation with an impact on heart disease may revolutionize the field of anesthesiology and could involve a change in paradigm in the induction of anesthesia and the use of hypnotics, which would ultimately be used as a perioperative therapeutic tool [26].

## 3. Enzyme Mechanisms of Myocardial Conditioning

At the cellular level, reperfusion injury is characterized by the accumulation of H^+^ ions, an acidic environment, calcium overload, and the formation of reactive oxygen species (ROS). These characteristics result in the opening of the mitochondrial permeability transition pore (mPTP) that modifies the fluidity and rigidity of the inner mitochondrial membrane, influences electron transport, and exacerbates mitochondrial dysfunction. All this leads to a vicious cycle that causes cell death by releasing cytochrome C into the cytoplasm and activating the caspase system that induces apoptosis [27]. Cardiac conditioning mitigates these injuries by preparing the myocardium for ischemic insult through brief episodes of ischemia and reperfusion [27,28,29].

Halogenated anesthetics are among the most widely used hypnotics due to their safety profile and the predictability of their anesthetic effect, as well as having properties that influence cardiac conditioning. This cardiac-conditioning mechanism involves the activation of intracellular molecules and signaling pathways that finally result in a decrease in cellular oxygen demand. Numerous studies have examined the action mechanisms and its beneficial effects of the aforementioned anesthetics on the myocardium, administered during and after cardiac surgery, showing that mitochondria play a fundamental role in them. The myocardium is a tissue rich in mitochondria that comprises a plethora of functions at the crossroads of cell death or survival, which includes the maintenance of aerobic ATP production, redox signaling, calcium transport, and cell-death pathways [30].

Preconditioning and postconditioning triggers could have pathways in common for protecting the mitochondria, reducing inflammatory mediators, and preventing mitochondrial calcium overload. Some of the mediators of these effects are adenosine-triphosphate-sensitive potassium channels (K_ATP_), ROS, apoptosis cascade, nitric oxide (NO), and intracellular calcium overload [31,32,33].

The mitochondrial K_ATP_ channel is a critical determinant of mitochondrial respiration. Its opening induces the depolarization of the inner mitochondrial membrane, preserving mitochondrial volume and homeostasis. This attenuates excessive ROS generation and mitochondrial calcium accumulation that provides the optimal medium for ATP production and inhibits mPTP pore opening. This channel is a direct target of volatile anesthetics [34,35].

### 3.1. Anesthetic Preconditioning Induced by Volatile Anesthetics

Numerous subsequent studies have addressed anesthetic preconditioning effects on the myocardium using enflurane, isoflurane, sevoflurane, and desflurane, aiming to also elucidate the underlying mechanisms for such effects. Thus far, evidence suggests that anesthetic preconditioning shares fundamental characteristics with ischemic preconditioning, providing early and delayed windows of myocardial protection [35,36,37].

In addition, anesthetic and ischemic preconditioning share many of the same molecular processes involved in myocardial protection, such as G-protein-coupled cell membrane receptors, mediation via multiple protein kinases, and the opening of K_ATP_ channels [24,35,38].

Two main intracellular signal-transduction pathways, directing cardioprotection from cell-surface receptors to convergent targets in the mitochondria, have been proposed as models to explain preconditioning by volatile anesthetics: the reperfusion injury salvage kinases (RISK) pathway via G-protein-coupled cell-surface receptors and the survivor-activating factor-enhancement (SAFE) pathway [24].

The RISK pathway is composed of a group of kinases that confer cardioprotection when activated immediately before or at the time of reperfusion [36,39,40]. This pathway is considered the main pro-survival kinase cascade and is a combination of two parallel cascades: the phosphatidylinositol-4,5-bisphosphate 3-kinase (PI3K)/Akt pathway, which prevents pro-inflammatory and apoptotic events through the exchange of signals with nuclear factor kappa B (NF-kB) and glycogen synthase kinase 3β (GSK3β), and the MAP kinase 1 MEK1/ERK1/2 pathway [30]. In brief, cardioprotective signaling by volatile anesthetics is mainly initiated by G-protein-coupled cell-membrane receptors, which include β1- and β2-adrenergic receptors and adenosine-A1 receptor [35]. Volatile anesthetics prime the activation of the sarcolemma and mitochondria K_ATP_ channels, the putative end-effectors of preconditioning, by stimulation of adenosine receptors and subsequent activation of protein kinase C (PKC) and by increased formation of NO and ROS. Activated PKC acts as an amplifier of the preconditioning stimulus and stabilizes, by phosphorylation, the open state of the mitochondrial K_ATP_ channel (the main end-effector in anesthetic preconditioning) and the sarcolemma K_ATP_ channel. The opening of K_ATP_ channels ultimately elicits cytoprotection by decreasing cytosolic and mitochondrial Ca^2+^ overload, cell death, and improved myocardial survival. In the case of desflurane, stimulation is done by β-adrenergic receptors [41].

Activation of PI3K results in the phosphorylation of pyruvate dehydrogenase kinase 1 (PDK1), which activates Akt to subsequently recruit a wide range of pro-survival targets such as endothelial nitric oxide synthase (eNOS), PKC, and the antiapoptotic protein Bcl-2 [42,43,44]. Furthermore, it phosphorylates and inhibits GSK-3β, which is responsible for mPTP opening [45,46,47]. On the other hand, activation of ERK 1/2 has been proposed as a redundant mechanism by which downstream elements of the PI3K-Akt cascade may be stimulated to favorably modulate reperfusion injury [48].

It has been shown that short-term activation of these RISK kinases is protective, triggering downstream pro-survival pathways, whereas long-term activation is considered harmful due to their growth-inducing effects and induction of cardiac hypertrophy [41].

The importance of the RISK pathway activated by halogenated agents is based on three fundamental concepts: (1) the short-term activation of its kinases is protective; (2) activated at the time of early reperfusion, it produces cardioprotection; (3) it is considered a universal signaling cascade, a common pathway shared by most cardioprotective therapies [39,49].

On the other hand, halogenated drugs also play a fundamental role through the SAFE pathway [50,51]. This pathway has been described as an alternative RISK-independent cascade that mediates cardioprotective effects with the JAK/STAT system. It mainly involves tumor necrosis factor-alpha (TNF-α) and the signal transducer and activator transcription of mitochondrial STAT3, which, when translocating to the mitochondria, inhibits the opening of mPTPs and promotes cardiomyocyte survival [52,53]. In humans, several publications have emphasized STAT5, for the attenuation of cell death by apoptosis, playing a fundamental role in cardioprotection [54,55].

SAFE potentiates its effects through the JAK kinase, which is originally activated by several cytokines, including interleukin-6 (IL-6) and TNF-α. JAK will subsequently activate STAT, which will interact with NF-kB signaling that stimulates mitochondrial fusion by activation of the optic atrophy protein (OPA1). This cascade of signals potentially affects respiration and mitochondrial inflammation, providing a protective response [40]. TNF-α may also stimulate the SAFE pathway after binding to TNF Receptor 2, which activates STAT3. STAT3 in dimer form has been shown to increase the anti-apoptotic gene Bcl-2 and decrease the apoptotic gene Bax in response to a conditioning stimulus. While STAT3 is a transcription factor, many STAT3 effects in a pre- or postconditioning setting do not occur at transcription but are resultant of phosphorylation of various components, such as the phosphorylation and inactivation of the pro-apoptotic factor Bad.

Additionally, it will inhibit the apoptotic factor forkhead box protein O1 (FOXO-1) in the nucleus (family of transcription factors that induces the expression of apoptotic genes) [50,53]. While TNF-α was originally thought to contribute to reperfusion injury, it may paradoxically contribute to the cardioprotection induced by pre- and postconditioning [56].

As with the RISK pathway, chronic stimulation of the SAFE pathway is detrimental to the heart [56]. In experimental ischemic heart failure, excessive stimulation of TNF will contribute to the processes of inflammation, apoptosis, and cardiac remodeling (via TNF receptor 1) [57,58].

Both the RISK and SAFE pathways have close interactions with each other, and, in both, protection in mitochondria is transmitted by inhibition of the mPTP pore and the opening of the K_ATP_ channel [59]. An interrelation between the RISK and SAFE pathways has been described. For cardiomyocytes of mice deficient in STAT3 or with its inhibitor present, preconditioning strategies do not protect or activate components of the RISK pathway such as Akt and ERK. Similarly, inhibition of the RISK pathway with Akt or ERK inhibitors results in unprotected conditioning strategies and failure to activate STAT3 [58,59]. Indeed, ERK may also be a key player in the regulation of serine phosphorylation of mitochondrial STAT3 [60,61].

Some cardioprotective strategies may require activation of the RISK and SAFE pathways, while other strategies may require only one of the two pathways of protection [52].

Experimental investigations suggest that the ability of volatile anesthetics to protect the myocardium by anesthetic preconditioning significantly increases from isoflurane to sevoflurane to desflurane. Not only the type of volatile anesthetic but also the duration and frequency of exposure to the volatile anesthetic before ischemia have been shown to be of potential relevance in in-vitro experiments [35].

### 3.2. Anesthetic Post-Conditioning and Other Direct Cardioprotective Effects Induced by Volatile Anesthetics

The second window of protection appears after 24 to 48 h and lasts until 72 h. It involves increased expression of protective proteins, including PKCε, STAT, and NF-kB [38]. The main upregulated proteins involved in postconditioning include inducible nitric oxide synthase, cyclooxygenase-2, superoxide dismutase, aldose reductase, and heme oxygenase. It has also been suggested that activation of sarcolemma K_ATP_ channels, in conjunction with the mitochondrial K_ATP_ channel, alleviates cytosolic calcium overload in the second protective window [35,51].

Volatile anesthetics also provide postconditioning effects, with efficacies similar to those of preconditioning in terms of reducing infarct size, when given within the first 30 s of reperfusion [43]. The underlying mechanisms are similar to those of preconditioning and involve the activation of G-protein-coupled cell-membrane receptors and the stimulation of downstream pathways. As in preconditioning, the major pathways involved are the RISK and SAFE pathways [35].

It has been proven that both Desflurane and Sevoflurane decrease the expression of Bax, a pro-apoptotic protein, and improve the expression of Bcl-2; in addition, caspase 3 and 9 mediators of apoptosis were inactivated through the activation of PI3K and ERK1/2 [62]. Additionally, in mouse heart in vivo it has been shown that phosphorylation of the Bcl-2-associated death promoter (Bad) occurs through the activation of Pim-1 kinase, a proto-oncogene-mediating Akt activity in cardiomyocytes [62]. The global effect is antiapoptotic and cardioprotective, achieving the inhibition of mPTP opening [37].

Guerrero et al. have demonstrated that the molecular mechanism of the cardioprotective effect of Sevoflurane, in addition to varying some enzymatic molecules, also varies certain miRNAs [26]. Different key enzymes in the development of halogenated cardioprotection are related to their expression through gene modulation, among them, Akt, ERK 1⁄2, and the STAT group proteins, which are all related to the SAFE and RISK pathways described previously [26].

## 4. Anesthetic Modulation of Oxidative Stress in Heart Failure

Reactive Oxygen Species (ROS) are highly reactive oxygen-containing free radicals such as superoxide (O_2_^−^) and hydroxyl (OH^−^) and chemical elements such as hydrogen peroxide (H_2_O_2_) that could generate OH^−^ free radicals through the Fenton reaction or by combining with nitric oxide (NO) to form peroxynitrite (ONOO^−^). In addition, OH^−^ could arise from the exchange of electrons between O_2_^−^ and H_2_O_2_ through the Haber–Weiss reaction [63].

Under physiological conditions, ROS levels are strictly controlled by enzymatic or non-enzymatic antioxidant defense systems. The main enzymatic defense systems are superoxide dismutase (SOD), catalase (CAT), and glutathione peroxidase (GPx) [64]. The non-enzymatic defense systems can be endogenous, defined as those that can be synthesized by eukaryotic cells, or exogenous, which must be ingested with food. The most outstanding endogenous non-enzymatic systems are the reduced glutathione (GSH) [65] and the Coenzyme Q10 (CoQ10), or ubiquinone [66], and the exogenous ones are Vitamin C (Vit C), or ascorbic acid (AA) [67], and Vitamin E (Vit E) [68], in addition to flavonoids, beta-carotene, and lipoic acid.

An imbalance between ROS and antioxidant defense systems is called “Oxidative Stress” and may induce oxidative damage at the DNA level, plasmatic membrane, proteins, and other cellular macromolecules.

Different levels have been established to define the intensity of oxidative stress. The lowest level has been identified as physiological oxidative stress or eustress, which affects products involved in cellular metabolism, and the maximum oxidative level has been identified as distress, which induce cellular toxicity through the activation of cellular antioxidant defense processes [69,70].

### 4.1. Oxidative Stress in Cardiomyocytes

In the heart, ROS contribute significantly to cellular homeostasis through the regulation of processes such as cell proliferation, differentiation, and excitation–contraction coupling [71]. When ROS generation exceeds the capacity of the antioxidant defense mechanisms or when different antioxidant enzymes are impaired, oxidative stress induces cellular disorders at the lipid, protein, and DNA levels; damage at the molecular level; and, finally, heart failure [72], myocardial remodeling with contractile dysfunction, and structural abnormalities of cardiac tissue [63].

Moreover, in the heart, the oxidative stress may be induced by cardiac hypoxia. In this situation, the oxygen concentration becomes a limiting factor for normal cellular activity such as ATP production, and it is associated with cardiovascular damage such as myocardial infarction, stroke, peripheral arterial disease, renal ischemia, and ischemia-reperfusion. The myocardium, under a hypoxia condition, increases blood oxygen extraction, resulting in an important coronary arteriovenous alteration such as the reduction in or interruption of coronary blood flow, named ischemic hypoxia [72], or the reduction in partial oxygen pressure (PO_2_) in arterial blood, named cardiac hypoxia [71]. Oxidative stress is one of the most important pathways that trigger both pathological situations generated by the hypoxic state of the myocardium.

### 4.2. Main Sources of ROS in the Heart and Their Pathological Action

The main sources of ROS identified in the heart are the following:−The mitochondrial respiratory chain:

The main mediators of ROS in mitochondria are the complexes I and III. Both complexes are responsible for most of the ROS released by mitochondria at the cardiovascular level, under physiological and pathological conditions [73,74]. In studies conducted in different animal models, it has been observed that modifications in the oxidative function of mitochondria reduce cardiac aging, protect against cardiac damage, and prevent left-ventricular remodeling [75,76].
−The enzyme xanthine oxide-reductase (XOR):

It is a homodimer of 30 kDa [77]. This enzyme is normally expressed in its dehydrogenase form (XDH), but under inflammatory conditions, it changes its reductase form to oxidase (XO). Both forms are responsible for the oxidation of xanthine to uric acid, favoring a flow of electrons destined for the NAD^+^ reduction to NADH in the case of the XDH isoform, or oxygen molecules’ reduction to H_2_O_2_ and O_2_^–^ in the case of the XO isoform [78]. Minhas et al. have shown that the XOR enzyme is the main source of ROS generation in the heart, and its positive regulation contributes to cardiac hypertrophy [79]. In addition, chronic inhibition of XO prevents oxidation of myofibrillar proteins, preserving cardiac function [80].
−The enzyme nitric oxide synthase (NOS):

This enzyme belongs to an enzymes group that catalyze the production of NO and citrulline from oxygen and L-arginine as substrates. Uncoupled NOS generates more ROS and less NO, modifying the nitroso–redox balance and causing adverse consequences in the cardiovascular system, while playing a key role in ischemia/reperfusion injury, cardiac hypertrophy, and cardiac remodeling [81]. Conversely, increased NO bioavailability may be considered as one of the universal mechanisms for cardiovascular protection against cardiac impairment. In the myocardium, three NOS isoforms are expressed: endothelial NOS (eNOS or NOS3), neuronal NOS (nNOS or NOS1), and inducible NOS (iNOS or NOS2) [82]. eNOS is expressed in coronary arteries, in endothelial cells of the endocardium, in cardiac-impulse-conducting tissue, and in cardiomyocytes [83]. Myocardial nNOS is preferentially in the sarcoplasmic reticulum. It has been suggested that nNOS-derived NO may inhibit Ca^2+^ influx through L-type Ca^2+^ channels and stimulate Ca^2+^ re-uptake in the sarcoplasmic reticulum by promoting phospholamban (PLN) phosphorylation. The nNOS-derived NO may also modulate the inotropic response to β-adrenergic stimulation and inhibit XOR activity, thereby limiting myocardial oxidative stress and, indirectly, increasing NO availability within the myocardium [81]. Finally, NO derived from iNOS isoform is considered to have detrimental effects on the myocardium. Indeed, upregulation of iNOS by IL-1β and IFN-γ cytokine increased secretion and has been shown to induce apoptosis in neonatal rat cardiomyocytes. In addition, the iNOS myocardial overexpression in mice showed cardiac fibrosis, cardiomyocyte death, cardiac hypertrophy, and dilatation [84].
−The NADPH-oxidase (Nicotinamide Adenine Dinucleotide Phosphate) (Nox) system:

NADPH oxidases (Noxs) are a family of seven plasma membrane enzymes and represent the main sources of ROS in the cardiovascular system [84]. They catalyze the reduction of molecular oxygen to O_2_^–^ using NADPH as an electron donor. One of them, Nox2, is abundantly expressed in cardiomyocytes, endothelial cells, and fibroblasts. It is a sarcolemma enzyme that is activated by multiple stimuli such as angiotensin-II (Ang-II), endothelin-1 (ET-1), TNF-α, growth factors, cytokines, and mechanical forces [84,85,86]. Another enzyme such as Nox4 is continuously expressed in endothelial cells, cardiovascular myocytes, and fibroblasts and increases its expression in damaged cardiac cells [87,88].
−Cytochrome P450 (CYPs) oxidase enzyme:

CYP isoform 2E1 (CYP2E1) is in the endoplasmic-reticulum membrane and is the most active CYPs in ROS production [89]. The expression level of CYP2E1 is significantly increased in human-heart tissues under ischemia and is directly involved in the pathogenesis of dilated cardiomyopathy. Its expression is associated with increased expression of oxidative stress markers and apoptotic processes in cardiomyocytes [90].
−The enzyme monoamine oxidase (MAO):

It is an enzyme in the external mitochondrial membrane. There are two isoforms: MAO-A and MAO-B. Both isoforms participate in the regulation of metabolism or degradation of catecholamines and other biogenic amines in mammals. Both are expressed at equivalent levels in the human heart [91]. MAO expression and its ability to produce ROS increase with age and are associated with chronic damage. In addition, MAO-A generation due to oxidative stress triggers p53 activation and impairs lysosome function. A genetic deletion of MAO-B has been shown to protect against oxidative stress, apoptosis, and ventricular dysfunction [92].

### 4.3. Major Cardiac Antioxidant Systems

Maintaining a balance between oxidants and antioxidants protects healthy organisms from the damaging effects caused by free radicals. The continuous generation of free radicals in eukaryotic organisms must be compensated by an equivalent rate of antioxidant substances [71]. Focusing on the heart, the three cell types of the myocardium (cardiac myocytes, fibroblasts, and endothelial cells) [62] possess the most important antioxidant systems [72], which are:−Superoxide dismutase (SOD):

SOD is a metalloenzyme that transforms O_2_^–^ into H_2_O_2_ and prevents the production of ONOO^–^ by blocking the oxidative inactivation of NO, which would cause important pathological consequences in the cardiovascular system [93].
−Catalase (CAT):

CAT is a tetrameric antioxidant enzyme that catalyzes the hydrolysis of H_2_O_2_ into oxygen and water. CAT is widely distributed in the peroxisomes of the cell cytoplasm when H_2_O_2_ concentrations increase, due to an inflammatory reaction [87,93].
−The enzyme glutathione peroxidase (GPx):

It is a cytosolic enzyme that also catalyzes the hydrolysis of H_2_O_2_ into oxygen and water and even the conversion of peroxide radicals into alcohols and oxygen. To date, there are eight different isoforms of GPx. The GPx-1 isoform is the most common. This isoform is in the cytoplasm and in the mitochondria of endothelial cells of the heart where it has been shown to participate as a cardiac protection mechanism [94].

### 4.4. miRNAs as Therapeutic Regulators of Oxidative Stress in the Heart

MicroRNAs (microRNAs) are composed of twenty-two naturally occurring nucleotides that control gene expression by pairing with specific messenger RNAs, preventing translation, or increasing degradation of the target messenger RNA (mRNA).

Currently, miRNAs are being identified as new therapeutic biomarker candidates for different pathologies, including cardiovascular diseases [95]. As cited above, miRNAs are one of the most important subjects of study in the field of cardiac-impairment diagnostics and therapeutics [24]. The preliminary results recently published by our group show several miRNAs as mediators of cardioprotection in patients who received sevoflurane as a halogenated agent in cardiac surgery. In these patients, we have observed variations in the expression of miRNAs associated with better prognosis of ischemic heart disease. These miRNAs are associated with the activation of mediators of anesthetic-induced pre- and post-conditioning, as well as cell apoptosis reduction and caspase and TNF-alpha concentrations decreasing [26].

Recent publications have described the miRNAs’ crucial role relating to cardiac diseases, such as miR-1. This miRNA is one of the most abundant and specific miRNAs in cardiac and skeletal muscle. It is an important regulator of cardiomyocyte growth in the adult heart, as well as a pro-apoptotic factor in myocardial ischemia, related to diseases such as hypertrophy, myocardial infarction, and cardiac arrhythmias. In addition, it can be used as a biomarker of myocardial infarction [96].

In addition, it has been shown that the expression of certain miRNAs is modified after the administration of antioxidant compounds, showing a protective mechanism against cardiovascular damage [97]. To date, according to published studies, miRNAs can be considered as potential targets and/or stimulators of pathways related to oxidative stress [98] and cardio-protection [26].

### 4.5. Role of Halogenated Agents during Cardiac Surgery

Oxygen administration is particularly relevant during and after cardiac surgery with extracorporeal circulation. High oxygen concentrations are administered with the intention of preventing cellular hypoxia in patients undergoing surgery under general anesthesia and in those with acute or critical illness. However, excess O_2_, or hyperoxia, is also known to be detrimental [99,100].

When ROS formation overcomes the barrier of antioxidant defense systems, the toxicity generated may induce oxidative stress through three different pathways: by excess feeding of the respiratory chain and the consequent mitochondrial uncoupling, by increasing ROS reactions with NO and the consequent generation of cytotoxic reactive nitrogen species, or by lipid peroxidation, compromising the cell membrane stability and, therefore, its functionality.

On the other hand, this generated oxidative stress may activate antioxidant defense mechanisms through positive feedback aimed to compensate ROS reactivity, detoxify prooxidants, and repair damage [101].

The high administration of O_2_ for induction of anesthesia during a surgical procedure could generate a pathological state of hyperoxia. Isoflurane (2-chloro-2-(difluoromethoxy)-1,1,1-trifluoroethane) and sevoflurane (fluoromethyl-2,2,2-trifluoro-1-(trifluoromethyl) ethyl ether) are the most used volatile anesthetics in clinical practice providing unconsciousness and analgesia [102]. The toxicity and beneficial effects of these drugs have been widely studied as well as their effect on oxidative stress, all of which are closely related to the prognosis of surgery.

The relationship of both drugs with oxidative stress and ROS production has been analyzed in various animal models of heart failure. Regarding oxidative stress, it has been demonstrated that in states of oxygen-concentration imbalance, such as hypoxia, isoflurane and sevoflurane have a protective effect on ventricular myocytes, reduce the expression of inflammatory factors and markers of oxidative damage, increase the expression of antioxidant enzymes such as superoxide dismutase and catalase, regulate the expression of apoptosis-related genes, and reduce oxidative stress and nitric oxide levels through the ROS and NOS levels’ modulation. Regarding the studies carried out to relate both drugs to ROS production, paradoxically. it has been observed that they may be involved in the beneficial effects of volatile anesthetics used in preconditioning [103,104].

Clinical studies have also been conducted to demonstrate the cardioprotective effect of both halogenated drugs, and it has been observed that they do not affect the cytotoxicity nor do they produce cell damage at the DNA level. In addition, both anesthetics are linked to increased activity of antioxidant enzyme defense systems and do not trigger oxidative damage processes in the intervened patient or DNA oxidation. These beneficial effects of halogenated drugs improve the clinical outcomes of patients undergoing cardiac revascularization surgery due to their cardioprotective effect induced through different mechanisms such as modulation of G-protein-coupled receptors, intracellular signaling pathways, gene expression, potassium channels, and mitochondrial function. In addition, administration of volatile anesthetics has been shown to reduce biomarkers of myocardial damage and short-term mortality after cardiac revascularization surgery [36,105,106]. Dharmalingam et al. [107] recently examined the relationship between volatile anesthetic administration and oxidative stress in patients undergoing cardiac revascularization surgery and concluded that preconditioning with the volatile anesthetics isoflurane and sevoflurane prevents oxidative and nitrosative stress during cardiac-revascularization surgery. Between these two halogenated agents, isoflurane provides better protection during the period before cardiopulmonary bypass, whereas sevoflurane provides protection during the periods before and after cardiopulmonary bypass. As cited above, we have demonstrated that the use of sevoflurane during the operative and postoperative process increases the overexpression of enzymes that reduce myocardial damage [24].

On the other hand, some published studies have questioned the beneficial effect of volatile anesthetics. Recently, Landoni et al. [17] have carried out a multicenter, randomized, blinded, and conftrolled clinical trial in which they observed that the use of volatile anesthetics during cardiac-revascularization surgery reduces short-term mortality in patients who underwent surgery; however, they have not observed differences regarding patients who received intravenous anesthesia. In this study, no study to determine the relationship between the administration of both halogenated drugs and oxidative stress was performed.

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
