# Peer review of "Heart Failure after Cardiac Surgery: The Role of Halogenated Agents, Myocardial Conditioning and Oxidative Stress"

_ijms, 2022, doi:10.3390/ijms23031360_

Round 1
Reviewer 1 Report
This manuscript will need to revise a few points.
1 Introduction
I didn't understand the hypothesis of this study. Please specify your hypothesis.
2.Study design and results
I think that the reproducibility of this study needs to be clearly stated. If it cannot be stated, it should be stated as the limit of research.
3. There is a lot of literature consideration about the discussion. Please consider based on this result. Minimize citations in the literature.
Reviewer 2 Report
The present review by Guerrero-Orriach, Carmona Luque and Gonzales Alvarez focuses on the impact of halogenated agents on cardiac dysfunction and report various mechanisms of pre- and postconditioning as well as their personal experience.
The study is well written and supports the current existing data situation.
I have the following comments:
- It is not made clear if your review is pointed to patients undergoing myocardial revascularization surgery or cardiac surgery in general. Please specify in the introduction section.
- The review mentions positive results on the mortality, however it is not made clear, if this especially applies to short- or long-term-mortality, if seen fit I would suggest clarifying.
- Consider also citing the meta-analysis "Volatile Anesthetics versus Propofol for Cardiac Surgery with Cardiopulmonary Bypass: Meta-analysis of Randomized Trials" since the results very much seem to be in line with your review.
Reviewer 3 Report
The manuscript is suitable for publication, while the Authors should be commended for their work.
